# The Role of Palliative Care in the Cardiac Intensive Care Unit

**DOI:** 10.3390/healthcare7010030

**Published:** 2019-02-19

**Authors:** Massimo Romano’

**Affiliations:** Centro Universitario Interdipartimentale di Ricerca in Cure Palliative, Università di Milano, via Ripamonti 428, 20136 Milano, Italy; max.romano51@gmail.com; Tel.: +39-3488131566

**Keywords:** intensive care unit, cardiac intensive care unit (CICU), palliative care, end of life care, withdrawing treatment, withholding treatment, heart failure

## Abstract

In the last few years, important changes have occurred in the clinical and epidemiological characteristics of patients that were admitted to cardiac intensive care units (CICU). Care has shifted from acute coronary syndrome patients towards elderly patients, with a high prevalence of non-ischemic cardiovascular diseases and a high burden of non-cardiovascular comorbid conditions: both increase the susceptibility of patients to developing life-threatening critical conditions. These conditions are associated with a significant symptom burden and mortality rate and an increased length of stay. In this context, palliative care programs, including withholding/withdrawing life support treatments or the deactivation of implanted cardiac devices, are frequently needed, according to the specific guidelines of scientific societies. However, the implementation of these recommendations in clinical practice is still inconsistent. In this review, we analyze the reasons for this gap and the main cultural changes that are required to improve the care of patients with advanced illness.

## 1. Introduction

Worldwide, the number of patients who die in hospital is high, albeit with considerable differences between countries [1,2]. The median rate is 54%, with a maximum of 78% in Japan and a minimum of 20% in China [2].

In the United States (US), 40% of people die in hospital and about half (22% of the total population) die in an Intensive Care Unit (ICU), after having spent almost their whole hospital stay there [3].

Teno et al. [4] confirmed the tendency to admit patients in the last stages of life to the ICU. They found an increase in such accesses in the last month of life for Medicare beneficiaries aged >66 years, from 24.3% in 2000 to 29.2% in 2009. An additional finding was an increase in hospice admittance at the time of death from 21.6% in 2000 to 42.2% in 2009. Forty percent of patients who died in a hospice in the previous month had been admitted to an ICU.

In the US, acute care hospital beds have been reduced [5,6], in favour of increasing hospital beds in ICUs; this has resulted in the greater use of ICU beds, which are very expensive [6].

The increase in deaths in hospital and the use of ICUs beds is largely due to the increase in mean age, predominantly in the >85 years age group [6,7].

## 2. Definition and Role of an Intensive Care Unit

According to the World Federation of Societies of Intensive and Critical Care Medicine [8] an ICU is “an organized system for the provision of care to critically ill patients that provides intensive and specialized medical and nursing care, an enhanced capacity for monitoring, and multiple modalities of physiologic organ support to sustain life during a period of acute organ system insufficiency”.

Three levels of ICU are defined according to the type of hospital in which they are located, their case-mix, the number and training of their medical and nursing staff, and the available technological support [8].

The need to periodically redefine the level and the objectives of ICUs derives from the changes in the types of patients, technological evolution, and the improvement of specialization of healthcare staff.

ICUs are places where patients with serious diseases are treated using advanced technology, with the intention of achieving stable clinical conditions. However, the mortality rates are still high [9].

Both North American [10] and European [11] Scientific Societies have published documents regarding Cardiac Intensive Care Units (CICUs). Both documents, besides defining CICU levels that are based on the expertise of healthcare staff and the available technological support and discussing novel organizational solutions, focus on the progressive evolution of CICUs starting from the original Coronary Care Units (CCUs). These were created 50 years ago for the diagnosis and treatment of arrhythmic complications of acute myocardial infarction [12]. They have become true specialist ICUs, following substantial changes in healthcare services, culture, and education, both in terms of training and certification [10,11].

## 3. Changes of Data in the Modern Cardiac Intensive Care Units

Katz et al. [13] were the first to report the results that were derived from the analysis of the Duke University database related to more than 29,000 patients who were consecutively admitted to the CCU between 1989 and 2006. No significant reduction in mortality was recorded, with stability both in the CCU (7.4%) and in the hospital (11.4%), whereas a significant increase in the complexity index, expressed as the Charlson score, was noted. Moreover, the incidence of acute myocardial infarction with ST segment elevation (ST segment Elevation Myocardial Infarction—STEMI) decreased, whereas acute myocardial infarction without ST segment elevation (Non ST segment Elevation Myocardial Infarction—NSTEMI), heart failure, and cardiogenic shock increased. The use of non-cardiac procedures, such as mechanical ventilation or central venous catheterization, increased over time.

The most significant information is related to the increase in severe non-cardiovascular conditions, such as septic shock/sepsis, acute or chronic renal failure, acute respiratory failure, and pneumonia.

Mortality data and survival predictors in the CCU are reported less in the literature than those that are related to general ICUs.

Ratcliffe et al. reported, in a multicentre trial, variable mortality rates of about 5.6–7.4% (range 2.2–9.2%) [14]. In 2011, acute coronary syndromes still accounted for most admittances (57%).

The survival was worse in patients resuscitated after cardiac arrest in hospital, sepsis, respiratory, and heart failure.

The prognosis of patients was poor when they had been submitted to aggressive treatments, such as defibrillation after cardiac arrest that occurred in the CCU, invasive right heart hemodynamic monitoring, mechanical ventilation inotropic or vasopressor support, emergency dialysis, or the positioning of an intra-aortic balloon pump. Within these clinical settings mortality in the CCU exceeded 10% and the main causes of death were cardiogenic shock, respiratory failure, multi-organ failure, and arterial hypotension [14].

About 30% of patients died within the first 24 h after admittance to a CCU.

In the decade 2003–2013, Medicare patients that were admitted to CICU amounted to 3.4 million: primary non-cardiac diagnoses increased from 38% to 51.7% [15]. They included infections (from 7.8% to 15.1%) and respiratory diseases (from 6 to 7.6%), whereas the prevalence of coronary artery disease decreased from 32.3% to 19%.

Additionally, the prevalence of cardiovascular and non-cardiovascular comorbidities increased, as follows: heart failure (from 13.9% to 34.4%), pulmonary vascular disease (from 1.2% to 7.1%), valvular heart disease (from 5.0% to 9.8%), and renal failure (from 7.1% to 19.6%). Usually, non-cardiac diagnoses are associated with higher mortality rates than heart diseases, with greater usage of non-invasive and mechanical ventilation and of central venous catheterization.

Similar data were reported by Jentzer et al.: 10,000 patients were consecutively admitted to a first level (most advanced) CICU at the Mayo Clinic from 2007 to 2015 [16]. In the study, the mortality rate amongst patients aged <70 years in the CICU was 4.1% vs. 7.5% amongst patients aged >70 years. The disease severity scores (APACHE III, APACHE IV, SOFA) and the frailty scores (Braden skin score) were predictors of higher mortality rates in patients >70 years, who were less likely to be discharged home than to a SNF (skilled nursing facility) [16].

In the same group of patients, the daily changes in the SOFA score (which analyses simple parameters relating to cardiovascular, respiratory, renal, and neurological functions, as well as coagulation) were significantly related to the mortality rate in the CICU and in hospital. This supports the concept that comorbidities and non-cardiovascular diseases have a negative impact on prognosis [17].

Finally, also in an observational study in a first level CICU at Harvard [18], non-cardiovascular diseases were common and in-hospital mortality was 17.6%. Of note, 36% of patients underwent mechanical ventilation, 45% were given inotropic agents, vasopressors, or vasodilators, and 7.6% underwent dialysis. What has changed in the CICUs of the third millennium?

First of all, as already explained above, the epidemiology of the patients that were admitted to CICUs has changed. This begs the question: when is admittance to a CICU indicated and cost-effective?

It is not easy to provide an answer, especially because, to date, Scientific Societies of Cardiology have not issued guidelines that set general criteria for admittance to a CICU.

Epidemiological snapshots of the current CICUs are provided in the literature, especially focusing on the new skills that cardiologists require, which resemble those of Critical Care Specialists [10,13].

Since the CICU has changed, its objectives should change too and become similar to those of traditional ICUs. Consequently, the objectives of treatment in critically ill patients in CICUs are more related to the support of cardiovascular, respiratory, and renal functions, until the underlying disorder or acute conditions resolve. In these cases, the objective is to gain time so that the underlying disease can be treated. Technological developments offer physicians a broad means designed to support to patients with life-threatening medical conditions. Sometimes, however, the conditions are incurable and this makes the replacement of vital functions progressively ineffective. Consequently, also in CICUs, the success of intensive care should be measured not only in terms of survival, but also in terms of the residual quality of life, quality of death, and quality of human relationships that are involved in each death [19].

Little attention is also paid to another important aspect: once patients have been admitted to the CICU, they may often receive treatment that can be considered to be futile [20,21], aggressive, and invasive. These treatments are to be considered excessive when the end of life is approaching and they are not consistent with the wishes and expectations of the patients and their families [22].

These considerations are at the core of decisions to withdraw/withhold treatments, which are designed to implement palliative care, instead of giving intensive care at the end of the life of a patient with chronic, worsening disease [23].

## 4. Introducing Palliative Care in the CICU

Discussing palliative care (PC) in the CICU may appear to be an oxymoron, a contradiction in terms.

This would be true if PC care were considered as end of life care. The definition of PC as end-of-life care delays its implementation, thus reducing its efficacy.

The definition and purpose of PC, according to the World Health Organization (WHO) [24], are based on particular and well-defined features: “Palliative care is an approach that improves the quality of life of patients and their families facing the problem that is associated with life-threatening illness, through the prevention and relief of suffering by means of early identification and impeccable assessment and the treatment of pain and other problems: physical, psychosocial, and spiritual. Palliative care:provides relief from pain and other distressing symptoms;affirms life and regards dying as a normal process;intends neither to hasten or postpone death;integrates the psychological and spiritual aspects of patient care;offers a support system to help patients live as actively as possible until death;offers a support system to help the family cope during the patients illness and in their own bereavement;uses a team approach to address the needs of patients and their families, including bereavement counselling, if indicated;will enhance quality of life, and may also positively influence the course of illness; and,is applicable early in the course of illness, in conjunction with other therapies that are intended to prolong life, such as chemotherapy or radiation therapy, and includes those investigations that are needed to better understand and manage distressing clinical complications”.

At this stage, it is important to define the key expertise that is required for PC [25]. Primary expertise is required for all healthcare professionals who attend to patients with serious illnesses. These professionals should have knowledge and basic skills in PC. They should be able to identify symptoms (especially pain, anxiety, and depression) and offer basic treatment for them, as well as discuss the objectives of treatment, patient preferences, prognosis, and advance directives that are related to end-of-life-choices and the possible need for cardiopulmonary resuscitation.

The specialist level includes healthcare professionals who are certified in the field of PC and the attending team consults them. Consultations regard the treatment of refractory pain or dyspnea and the more complex forms of anxiety, depression, existential issues, and the assistance required for the resolution of potential conflicts between family members and the attending team.

A typical example of teamwork is the case of a 78-year-old patient, with previous aortocoronary by-pass and ICD implantation, with advanced heart failure. Many co-morbidities were present: diabetes, macrocitic anemia, polyneuropathy, with severe pain of the legs. This is a typical patient in whom a PC intervention has made pain control possible, during the last weeks of life.

The patients admitted or who will potentially be admitted to a CICU who can express palliative needs range from those with acute cardiovascular disease (who require palliative support in the stages when decisions have to be made regarding the therapeutic options available) to those who have chronic cardiovascular disease, who have had acute episodes in the past and experience the progressive deterioration of general clinical conditions, particularly at an advanced age, together with multiple comorbidities and pulmonary infections or sepsis. The patient with advanced heart failure (HF) accounts for most of them [26].

The recommendations in the literature [27] suggest that, for the patient in ICU (but the principle can be applied to other clinical settings as well), the traditional concept of starting PC near end-of-life, when “active” treatment is gradually withdrawn is obsolete. Early holistic palliative intervention that is designed to improve communication with the patient and family members is proposed [28].

Historically, the palliative care (PC) model, which is borrowed from oncology, consisted of the introduction of a PC program when the prognosis would become unfavourable within six months. The model proved not to be effective, because PC was often implemented late, when death was approaching. The patient with advanced HF generally has a notable burden of multiple, severe, physical, and psychosocial symptoms that occur simultaneously [28]. They affect quality of life considerably and cardiologists do not pay attention to them specifically, since they focus on the issues that are related to their field of expertise, especially in the acute phases of the disease.

Scientific Societies recommend a shift from interventions based on prognosis (which, as already said, is difficult to predict in the HF patient) to a model based on symptoms and quality of life [28].

The most modern model of PC starts with early introduction, while disease-modifying therapy is maintained (simultaneous care) and it is to be gradually withdrawn according to the progression of the underlying disease and the patient’s wishes. Continuing to refer to advanced HF, there is a time when decisions regard the full activation of PC or a program of mechanical circulatory support or cardiac transplantation should be taken [28].

At every stage of the course of the HF patient, additional in-depth discussions may be required with the patient, the family, and the caregivers, regarding the stage of disease, change in prognosis, and shared reassessment of therapeutic options. The discussion should be periodically updated, when also considering that HF, as with other organ failures, unlike cancer, sudden death, or frailty, often results in multiple exacerbations (that nearly always result in hospitalization) alternated with periods of prolonged recovery, whereas in other cases its course consists of relentless general deterioration (Figure 1).

The prognosis is therefore difficult to formulate and it is extremely variable; the rate of decline is unpredictable.

For example, a 71-year-old man that was affected by non-ischemic dilated cardiomyopathy with very low ejection fraction, advanced renal failure, with implanted biventricular defibrillator, was fully informed about the worse short-term prognosis and then decided to refuse any treatment and decide to go back home, where he died four days later.

The prognosis of the HF patient, despite improvements over the last few years thanks to pharmacological and non-pharmacological treatments, is still unfavorable (the mortality rate is still just under 50% after five years) [28] and similar to the mortality rate of some cancers; the mortality rate within one year after the diagnosis exceeds the mortality rate of all cancers except lung cancer [29].

Decisions regarding the available therapeutic options should be shared, and care should be planned, anticipating the decisions that will have to be taken in the final stage of life (Advance Care Planning—ACP).

ACP enables patients to define their preferences and expectations from therapeutic options; it is a process that helps adult patients of any age or any health conditions to understand and share their personal values, life objectives, and preferences in terms of future healthcare from a physical, psychological, social, and spiritual point of view [30].

A large majority of patients, if they are adequately informed, express the desire to discuss ACP with their attending physician. They manage to do so only in a minority of cases (15%), and even fewer of them (10%) believe that the doctor has understood their wishes [26].

The advance directives (ADs) are an essential part of ACP; ADs were defined as the presence of a living will, do-not-resuscitate order, do-not-hospitalize order, medication restriction, or feeding and hydration restriction, in the case of future loss of competence.

However, the intrinsic limits of ADs should be stressed if not included in the wider approach of ACP.

Advance directives presume an unrealistic control over the future.

It is not possible to fully predict acute exacerbations, making ADs difficult to adopt and sometimes misleading.

Finally, patient wishes may vary over time and according to the different phases of the diseases [31].

During the periodic discussions, aspects that are related to the main complications of HF should be dealt with in-depth, especially sudden death and, in patients with implanted devices, those that are related to their possible deactivation at the end of life [32,33].

The process of informing the patient and the family members can be rough. The patient and the family should receive exact information at the beginning of the disease, obviously a little at a time, according to their psychological, cultural, and social values.

## 5. Symptoms Evaluation and Control in the CICU

In the last weeks of life, major symptoms occur that may be disabling and always impair quality of life, with pain, dyspnoea, and depression being the most common. The higher their prevalence, the more frequent that admittance to the ICU is [4], where life sustaining treatment and aggressive medical therapies are carried out [34].

Studies on patients admitted to the ICU provided similar results [35].

Additionally, many patients suffering from advanced heart disease, especially HF, experience major symptoms, including mood disorders, insomnia, and gastrointestinal disturbances, such as nausea, vomiting, constipation, and anorexia.

The SUPPORT trial [36] and the Regional Study of Care for the Dying (RSCD) were the first two studies that assessed this issue [37].

In the former [36], of 263 HF patients, 65% were suffering from dyspnoea and 42% from severe pain during the last three days of life.

In the latter [37], a retrospective population study that was based on a sample of 675 English patients who died of cardiovascular disease in 1990, severe pain was reported by 50% of patients, dyspnoea by 43%, anxiety by 45%, and depression by 59%. These symptoms had sometimes been present for several months.

In the studies that assessed pain in patients with heart disease, the cause of the pain was not the heart disease, but it was mainly due to musculoskeletal disorders or was of neuropathic origin [38].

Table 1 reports the data relating to the main studies that assessed the prevalence of the most important symptoms that were reported by HF patients during their last six months of life [34,35,36,37,39,40,41].

In general, HF and cancer patients have the same high symptom burden, extent of depression, and reduced spiritual well-being (between 35% and 40% of patients), independently of their ejection fraction (<0.30 or >0.30). More than half of HF patients suffer from dyspnoea, pain, asthenia, and thirst [42].

Patients with advanced HF report a higher number of symptoms than patients with advanced cancer (13.2 vs. 8.6), are more depressed, and experience less spiritual well-being [41].

This means that HF patients have the same palliative needs as cancer patients.

Also in the ICU the symptom burden is high, especially in terms of depression, pain, dyspnoea, sleep disorders and thirst [43,44,45,46] and in 20% of cases this includes delirium. Patients with delirium have a higher mortality rate than those without (27% vs. 3%) [47].

An underestimated symptom, which is very common and responds poorly to treatment, is thirst, which occurs in the ICU in about 70% of patients. It is mainly due to fluid restriction, sympathetic nervous system, and renin-angiotensin-aldosterone system activation, pharmacological treatment, such as opioids (doses > 50 mg/daily), furosemide (>60 mg IV/daily), serotonin inhibitors, and to low levels of ionized calcium, besides ongoing mechanical ventilation [48,49].

Patients describe thirst as an important symptom, especially during fluid restriction on account of exacerbations of congestive heart failure. It is disabling and has impact on quality of life [49].

Identifying and treating symptoms other than those due to heart disease is important, since palliative treatment is additional therapy. It does not replace disease-modifying therapy and it does not apply only to terminal patients.

Symptoms are therefore severe, although it is not easy to measure their actual severity.

The measurement of symptoms in HF patients has not been standardized [50]. One of the tools that is most used is the ESAS (Edmonton Symptom Assessment Scale), which is a multidimensional scale that is easy to apply, which was developed to assess nine common symptoms in cancer patients: pain, tiredness, drowsiness, nausea, lack of appetite, depression, anxiety, shortness of breath, and wellbeing. A tenth symptom may be added optionally, according to specific clinical setting or patient request.

ESAS has also been validated also in heart failure patients [51].

Two specific and widely validated questionnaires can be used to measure the quality of life in HF patients: the Kansas City Cardiomyopathy Questionnaire (KCCQ) and the Minnesota Living with Heart Failure Questionnaire (MLHFQ).

KCCQ refers to the severity of physical restrictions, to the frequency of symptoms (dyspnea, swelling, fatigue, dyspnea, sleeping upright), to quality of life, and to restrictions in social functions. The lower the score is, the worse the quality of life is [50].

MLHFQ includes 21 physical and emotional items, as measured on a six-point Likert scale, representing different degrees of the effect on quality of life of HF patients: the higher the score, the worse the quality of life [50].

The score may vary from 0 (none) to 5 (very much).

Mood disorders can be measured by means of the Hospital Anxiety and Depression Scale (HADS). The HADS is a fourteen-item scale: seven items relate to anxiety and seven relate to depression. For each item, the score may vary from 0 to 3: A total score < 7 for both scales indicates non-cases, from 8 to 10 indicates mild or borderline cases, from 11 to 14 indicates moderate, and from 15 to 21 indicates severe cases [52].

Finally, it can be useful to use a global function rating scale, such as the Palliative Performance Scale (PPS), which analyses the residual functional abilities of the patient, which can be integrated with the Activity Daily Living (ADL) score as appropriate, and it has proved to be fairly closely correlated with KCCQ and NYHA Functional Classification [50].

The treatment of symptoms in patients with advanced HF (details provided in papers focusing on this issue [53]) should be instituted, also bearing in mind the potential adverse effects of the medications prescribed.

The use of morphine, which is still underused, and of opioids in general should be increased in order to achieve adequate control of pain and dyspnoea.

Finally, one should also take another underestimated aspect into consideration; about 30% of patients that were admitted to the CICU are malnourished and the rate of malnutrition increases to 50% in the elderly [53]. If malnutrition is not identified correctly, then it can affect the duration of CICU stay and its outcome, especially in patients with congestive heart failure [54].

## 6. Withholding/Withdrawing Treatments in the CICU

The difficulty in establishing an exact prognosis, especially of HF elderly patients with comorbidities, is a problem when the possible withholding/withdrawing life-sustaining treatments (LSTs) are discussed with patients (when competent), a health care proxy (if indicated), family members, and within the attending team. The key question, however, at the core of the issue is what decision to adopt in the absence of advance directives for patients who lack a decision-making capacity. The health care proxy should use what is known about the patient’s values, guided by substituted judgment. The last option, if the previous are absent, is the patient’s best interest.

In the most important chronic, progressive, and incurable non-oncological diseases, performance status is effectively complemented by the staging and evaluation of the underlying organ failure (heart, lungs, nervous system, kidneys).

There are three main functions to sustain: cardiac, respiratory, and renal function. The main interventions are the administration of inotropic agents, the use of an intra-aortic balloon pump, mechanical circulatory support (LVAD—Left Ventricular Assist Device), mechanical ventilation, and renal replacement therapies.

The LSTs include also artificial nutrition, antibiotics, blood products, and fluids.

Most of the deaths in ICU occur after withholding/withdrawing LSTs [55].

Italian data [23] collected in 84 general ICUs relating to 3793 patients who died or were discharged in pre-terminal conditions reported the decision to forgo life-sustaining treatments in 62% of cases.

Data that are related to withholding/withdrawing treatments in patients in critical conditions are reported only for ICU patients, both in Anglo-Saxon countries and in Italy, whereas none are available on CICUs.

The decision to withhold/withdraw LSTs, in specific clinical settings, is ethically appropriate. There is a general consensus that there are no ethical differences between these two practices, although intensive care physicians experience more psychological discomfort with withdrawing than with withholding life sustaining treatments [56].

Withholding/withdrawing LSTs is different when compared to euthanasia/assisted suicide; in the former cases, the death following the interventions is the natural consequence of the underlying disease. In the latter cases, the death follows an active intervention of the doctor, directly (euthanasia) or through the patient (assisted suicide) administering a lethal medication.

While there is a general agreement regarding withholding/withdrawing LSTs as being ethically and legally appropriate, euthanasia/assisted suicide in many countries are considered unethical and illegal.

The details of all the arguments related to withholding/withdrawing life-sustaining treatments are not the goals of this article. The reader is referred to more specific analyses [57].

Noteworthy is the issue of the deactivation of Implantable Cardioverter Defibrillators (ICDs) and Left Ventricular Assist Devices (LVAD) at the end of life.

ICDs are implanted for the primary or secondary prevention of sudden death [58]. Regarding the deactivation of ICDs, the issue is how to prevent frequent and painful interventions of the device without any substantial improvement in the duration and quality of residual life, at the end of life. A review on this subject [59] reports ICD intervention rates in the last weeks/hours of life of about 30%, with an arrhythmic storm in 25% of cases.

According to Sulmasy [60], some distinctions have to be made. Treatments of any kind may be subdivided into two types: “regulative” or “constitutive” therapies. The objectives of the former are to restore homeostatic balance in the body (e.g., antiarrhythmic and antipyretic agents and ICDs). The latter replace a function that the body can no longer provide for itself.

Constitutive therapies can, in turn, be subdivided into replacement or substitutive therapies. The former refer to technological interventions that are designed to become an integral part of the body and replace a physiological function completely (transplanted organ). The latter replace function in a non-physiological way (dialysis, mechanical ventilation, left ventricular assist devices) and they do not become an integral part of the patient’s body (Figure 2).

These distinctions can support the decision to deactivate a device or not. It is not ethical to “deactivate” a transplanted organ e.g., by administering potential lethal drugs to the organ, whereas it is possible to withdraw dialysis or mechanical ventilation, in particular, clinical settings, especially at the end of life [60].

ICDs are generally identifiable as a regulative therapy [60], since they meet the requirement of restoring cardiac rhythm to its normal balance when altered by a temporary anomaly. Based on this consideration, it is ethical to deactivate them.

From an ethical point of view, the deactivation of an ICD (an act that does not result in the immediate death of the patient) is a possible option. On the contrary, within specific cultural and legislative context in the different countries, the deactivation of a cardiac pacemaker (PM) in PM-dependent patients is not ethically acceptable, because deactivation would result in the immediate death of the patient. In the event of non-PM-dependent patients and in those undergoing biventricular pacing, deactivation does not result in immediate or short-term death, but it can worsen the quality of life, determining symptomatic bradycardia or hemodynamic deterioration.

Therefore, a distinction can be made between the deactivation of a PM, especially in PM-dependent patients, and of an ICD. From an ethical point of view, the deactivation of an ICD is a feasible option that is consistent with the palliative approach to end of life: death does not occur at once, it is due to the final progression of the underlying disease, does not introduce another disorder or does not induce death actively, even when this has been agreed with the patient. Therefore, it is not the same as euthanasia or assisted suicide. The European Consensus Document [61] provides behavioural guidelines, in particular: (a) the deactivation of an ICD should be the final result of a transparent and deliberate process, with full tracking and documentation (also in writing) of such a decision by the patient and physician and (b) the option of deactivating an ICD, in the event of worsening of general health conditions, should be discussed with the patient at the time of implantation and be an integral part of informed consent.

On account of the scarcity of organ donors and the progress of technology enabling better results, a growing number of patients with advanced HF underwent the implantation of Ventricular Assist Devices (VAD), either as a bridge to transplant (BTT) or as destination therapy (DT). These devices usually support only the left ventricle (LVAD). Some patients may experience LVAD-associated complications, such as bleeding, thromboses, infections, renal failure, and stroke and right HF in LVAD carriers [62]. Furthermore, other diseases may occur, especially in patients who receive VAD as DT. For these reasons, the patients (or their health care proxy if the patient is incompetent) may draw the conclusion that VAD is associated with more harm than benefit and then ask for its deactivation. An American study showed that LVAD was deactivated in 60.5% of patients before death [62]. In most cases (85.7%), the request for deactivation was made by the family, because the patient was incompetent. After the deactivation of LVAD, 89.4% of patients died within one hour and the maximum survival time amounted to 26 h. More than 87% of patients died in the ICU.

An ethical consensus exists that a competent patient (or his/her health care proxy) can ask for the withdrawal of life-sustaining treatment, when it has become ineffective or it is judged to worsen rather than improve quality of life [60]. The deactivation of a VAD can be perplexing, because VAD is a long-term continuous treatment that sustains an essential function that the body can no longer provide for itself and because deactivation inevitably results in death, often in a few minutes. It should be pointed out that VAD is not a replacement therapy, but it is classifiable as substitutive therapy (see Figure 2).

Therefore, its deactivation is ethically acceptable and it cannot be considered to be the same as assisted suicide and euthanasia, because the cause of death is related to the underlying disease, without the introduction of another disorder. Deactivation of VAD is also included in the guidelines of the International Society of Heart and Lung Transplantation on mechanical circulatory assist devices [63].

Mechanical ventilation, if present, should be stopped and an ICD (nearly all of these patients are carriers) should be deactivated at the same time as VAD deactivation. Finally, it is essential to not forget to deactivate alarms when the VAD is turned off to prevent the destabilizing sound of the alarm from intruding on a moment that is hoped to be of intimacy and peace for the patient and loved ones. The procedure varies according to the model and it should therefore be verified beforehand in every case.

The end of life of these patients is complicated and it also has a heavy impact on the caregiver, a key figure for these patients. A recent American study assessed the experience of caregivers in end of life situations [64]. A high degree of confusion regarding the end of life of VAD carriers and modalities of death emerged, as well as inadequate information from the attending healthcare team and the absence of a palliative care program.

Regarding palliative care, caregivers complained about the not unusual feeling of being abandoned by healthcare professionals that had been their reference and also serious concern regarding the poor level of knowledge that the palliative care team had on VAD. The latter issue is real and it is an almost hard obstacle for VAD carriers at the end of their life in a setting outside the hospital.

A final, but not insignificant, issue concerns palliative sedation, a specific infusion of drugs to obtain a reduction of the level of consciousness in the terminally ill patient to manage one or more refractory symptoms. The drugs are titrated to the effective dose: the most frequent refractory symptoms are dyspnea, pain, and delirium. Despite a long and persistent debate about some of the ethical aspects of palliative sedation, there is a wide agreement in considering palliative sedation different from euthanasia, because the goal is not hastening death, but reducing suffering at the end of life. Midazolam and propofol, which are sometimes associated with opioids if the pain is the refractory symptom, are the drugs that are infused at increasing doses until the best symptom control is achieved [65].

## 7. Ethical Issues of Palliative Care in the CICU

Formal education in end-of-life care is inadeguate in medical school and residency training.

Physician training is addressed to always fight death, even when this is no longer feasible; all available technologies should be used to reach this objective, independently of human, social, and economic costs. Certainly, in some circumstances, the temporary continuation of treatments are considered futile, but they may be worthwhile in a terminal patient if it enables the patient to have a few last moments of contact with family members. However, this becomes unacceptable if the objective is to meet the wishes of the family, especially if the decision may increase the patient’s suffering. The main objective of intensive care is the treatment of the underlying disease: this means that intensive care physicians, cardiologists included, have difficulty in considering that symptoms control and the assessment of patient needs, which are the cornerstone of palliative medicine, are important. This is essential for adequate management of patients with a poor prognosis or whose quality of life is expected to deteriorate. These patients will receive few clinical benefits from intensive care, in the face of associated high costs, both in terms of expenses and human suffering. The management of patients in critical conditions is influenced by the expectations of the patients and family members, which may be unrealistic on account of insufficient information and communication with the patient, and, overall, by current social and spiritual values, as well as personal beliefs.

Major technological evolution has occurred, especially in cardiology and it has significantly increased the odds of surviving very critical conditions. However, at the same time, the possibilities of technology have become more and more overrated, resulting in two potentially negative consequences: technological and treatment imperatives [66].

The former arises from doctor’s willingness to use a treatment because it is available, even if it is not indicated or is yet unproven, only to meet the emotional need to do something.

The latter derives from the wishes of the doctor, the patient or the family who feel obliged to give treatment that cannot provide clinical benefits, according to the logic of “doing everything we can”.

This means that aggressive and expensive treatments only prolong suffering and delays inevitable death, sometimes without any evidence from clinical trials [66].

There are two more fundamental elements regarding the decisions about the treatment of patient with advanced and severe disease: the dying process and the uncertainty.

Hilton and Bellomo defined the dying process as a meta-diagnosis, which was to be considered as well as the traditional diagnosis, in all patients with severe disease, those who are older, and those with co-morbidities [67].

This implies that a new competence should be implemented in intensive care: the competence of intensivist as “nosocomial thanatologist” [67].

The intensivist and cardiologist, should know the main aspects and the technological limits of LSTs, and may explain them to the patient, if competent, and to the family and the colleagues.

A cardiologist in the CICU may intervene in specific situations, such as in patients with advanced heart failure, frailty, and with worsening clinical conditions.

Which decisions should the cardiologist adopt: an aggressive treatment, technology-mediated or palliative approach and a symptom-oriented treatment?

These decisions are difficult and controversial, and they are characterized by uncertainty, the second element to consider when determining the best treatment for a specific patient, including their wishes.

The uncertainty concerns the diagnostic, prognostic, and therapeutic aspects.

The level of uncertainty is directly related to the degree of complexity of the patient, resulting from the underlying disease, the number of co-morbidities, and from the interactions of non-biological components (economic, social, cultural, and family status) [68].

Furthermore, there is evidence that admittance to the ICU is also influenced by bed availability: the higher the number of ICU beds available, the higher the probability that patients who will not benefit from ICU are admitted.

Their general conditions could frequently be either too poor or too good to obtain significant clinical advantages [69].

The high admission rate to ICU involves more use of invasive, expensive treatments without a decrease in mortality, in comparison to facilities where access to ICU is less frequent [70].

Patients with acute myocardial infarction or HF that were admitted to hospitals with the highest ICU or CICU admission rates showed lower quality of care, lower adherence to guidelines, and a higher 30-day mortality [71].

These considerations highlight the need to define CICU admittance criteria and the treatment protocols therein. Criteria should be laid down that are designed to ensure proportionality of care, which is based on balancing the appropriateness and burdensomeness of treatments (Figure 3) and adopting a correct bioethical approach (autonomy, beneficence, non-maleficence, justice). The proportionality of the treatment must be individualized, in each particular patient, with their clinical, social, and personal values and they must be assessed both by the physician and the patient, or the health proxy, if present, or the family if the patient is incompetent. The physician, on the basis of their experience and competence, will specify the appropriateness of the treatment. The concept of proportionality is strictly linked with the relationship between patient and doctor; otherwise, it could be only the rough application of therapeutic techniques [72].

## 8. Conclusions

The development of technology in cardiology has enabled a significant increase in the survival of patients with heart disease, who are at advanced clinical stages of disease and that have a higher burden of comorbidities that were not so frequent in clinical practice until a few years ago, as in patients with advanced HF.

Cardiologists operating in the CICU are untrained to deal with issues that are not part of their culture, such as irreversible disease and the needs of dying patients and their families. Cardiologists do not have experience in the care of patients in the last stages of life and in communicating with them.

The fragmentation of medicine has likely contributed to this situation, since it has shifted its focus away from the patient to the disease, splitting actions, responsibilities, and, above all, objectives. The inability to take care of patients, living in the illusion that they can cure them, relegates death to an adverse event, a statistic data, or an unnatural element that cannot be dealt with. The consequences of this underestimation can produce significant mistakes in terms of ethics, communication, and care.

Culture is the key. Therefore, Scientific Societies and above all university faculties should take on more responsibility for specific education. They should promote a program focusing on the patient’s needs rather than on the disease, including learning of communication with patients and family members [73,74,75].

## Figures and Tables

**Figure 1 healthcare-07-00030-f001:**
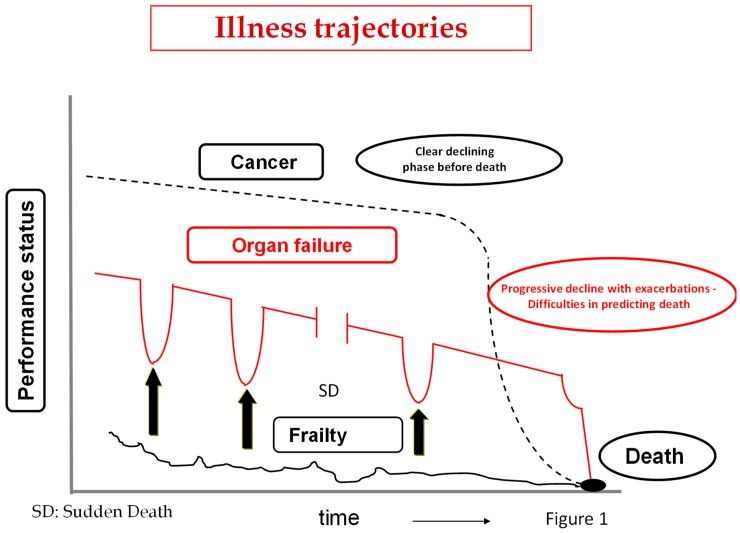
The arrows indicate exacerbation phases in organ failure associated with the risk of dying. The course of the disease may be stopped by sudden death (SD).

**Figure 2 healthcare-07-00030-f002:**
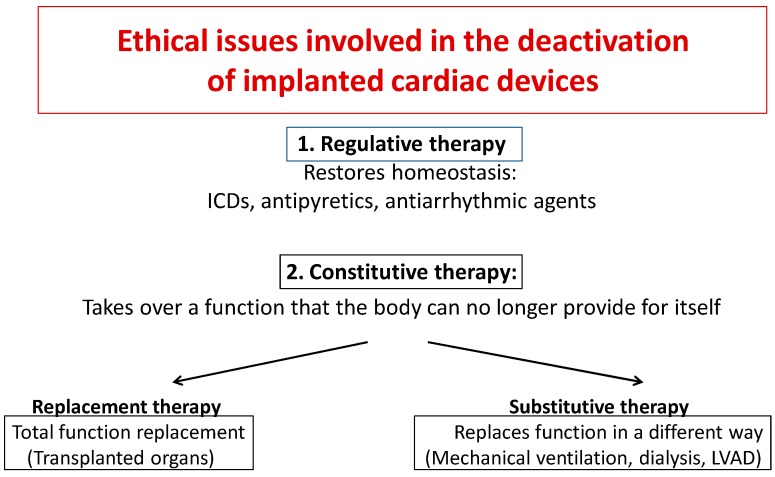
Modified from Sulmasy [60].

**Figure 3 healthcare-07-00030-f003:**
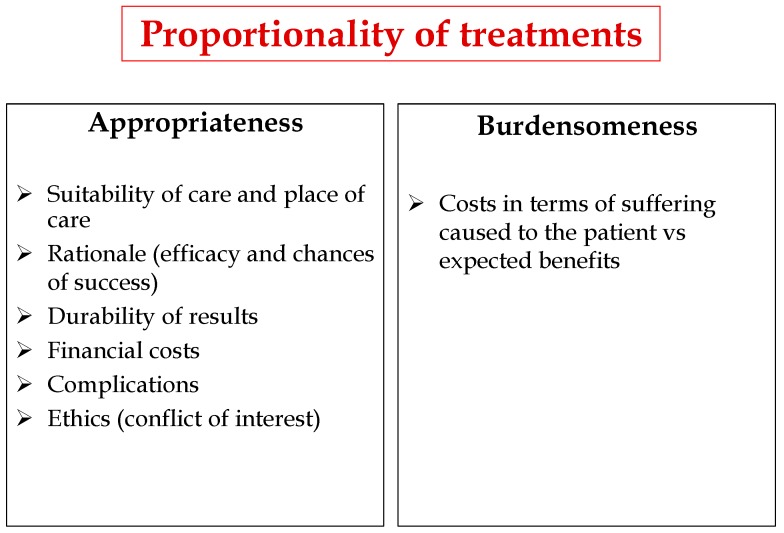
Factors affecting Proportionality of treatments.

**Table 1 healthcare-07-00030-t001:** Main symptoms prevalence at the end of life in patients with heart failure. The numbers close to the Author’s name are references numbers

	Dispnea	Pain	Depression	Fatigue	Anxiety	Nausea	Constipation
SUPPORT (36)	65%	42%					
RSCD (37)	43%	50%	59%		43%		
Nordgren (40)	88%	75%	9%	69%	49%	48%	42%
Singer (34)	75%	64%	63%	68%			
Nelson (35)		65%	80%	85%	80%	52%	
Levenson (39)	63%	41%	70%		70%		
Zambroski (41)	85%	57%	61%	81%		41%	26%

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
