# Peer review of "The Role of Palliative Care in the Cardiac Intensive Care Unit"

_healthcare, 2019, doi:10.3390/healthcare7010030_

Round 1
Reviewer 1 Report
Comment 1: The paper needs to give a better definition about the difference between withdrawal/withholding and euthanasia/assisted suicide.Both in the Ethical issues and in the withdrawal/withholding sections in my opinion a more focused-explanation about ethical principles, risk of ethical dilemmas is needed; a focus on relation physician-patient/family could be useful, also in the context of Advanced Care planning.
Comment 2: about the symptoms section. It could be very useful to give more details about the tools the author talks about.
Comment 3: About the issues concerning medical training in communication skills, ethical discussion skills and patients' needs skill : it's important to give some references from literature about the statement of inadequate skills of physician.
Author Response
The new text included after the comments is colored in green.
Comment
1: The paper needs to give a better definition about the difference
between withdrawal/withholding and euthanasia/assisted suicide.Both in
the Ethical issues and in the withdrawal/withholding sections in my
opinion a more focused-explanation about ethical principles, risk of
ethical dilemmas is needed; a focus on relation physician-patient/family
could be useful, also in the context of Advanced Care planning. Response: In the paragraph titled: Withholding/withdrawing treatments on the CICU
it was inserted a further consideration about the difference between
withholding/withdrawing treatments and euthanasia/assisted suicide. In
the same paragraph and in Ethics of palliative care in the CICU a new
text is proposed about Advance care Planning, Advance Directives and
ethical problems concerning clinical decisions and uncertainty in
medicine. Comment 2: about the symptoms section. It could be very useful to give more details about the tools the author talks about. Response: A more details about the tools of symptoms evaluation were added. Comment
3: About the issues concerning medical training in communication
skills, ethical discussion skills and patients' needs skill : it's
important to give some references from literature about the statement of
inadequate skills of physician.
Response: Some additional references were added.
Reviewer 2 Report
Role of palliative care for patients suffering end-stage/life-threatening heart diseases is extensively addressed in this review with a specific interest for clinical conditions that are frequently managed in intensive (general or cardiologic) care units. The Author also provides recommendations on when and how to deliver palliative care to patients admitted to the CICUs (for example HF patients), acknowledging a lack of clear guidelines issued by the scientific societies.
In my opinion the subchapters reporting on “epidemiology” and “changes of the CICUs of the third millennium”, are too long (with some repetitions) and should therefore be downsized, while the subchapter dedicated to “palliative care”, which is actually the core business of the entire review, should be better structured following the logic sequence of: 1. What is the correct definition of palliative care; 2 What is the purpose of complementing intensive care, 3. Role, advantages and main issues of the three phases of the “Advance care planning”, “symptom control” and “withholding/withdrawing life-sustaining treatments” at the end of life, supporting the theoretical concepts with the examples of real clinical situations (such as the paradigmatic heart failure management, that has been frequently mentioned)
Ethics related to the appropriate intensive care delivery is out of the purpose of the paper and should be probably restricted to few basic concepts, while ethics of palliative care is extensively addressed and should remain as it is. With regards to lines 337-340, where the Author claims the need of better defining CICU admittance criteria, it is in my opinion obscure the concept of “proportionality of treatments ” as well as table 2, both deserving further clarification
Author Response
Role of palliative care for patients suffering end-stage/life-threatening heart diseases is extensively addressed in this review with a specific interest for clinical conditions that are frequently managed in intensive (general or cardiologic) care units. The Author also provides recommendations on when and how to deliver palliative care to patients admitted to the CICUs (for example HF patients), acknowledging a lack of clear guidelines issued by the scientific societies.
Response: Thank you very much for your kind review of the manuscript. The new text included after the comments is colored in yellow and the deleted text is done.
In my opinion the subchapters reporting on “epidemiology” and “changes of the CICUs of the third millennium”, are too long (with some repetitions) and should therefore be downsized, while the subchapter dedicated to “palliative care”, which is actually the core business of the entire review, should be better structured following the logic sequence of: 1. What is the correct definition of palliative care; 2 What is the purpose of complementing intensive care, 3. Role, advantages and main issues of the three phases of the “Advance care planning”, “symptom control” and “withholding/withdrawing life-sustaining treatments” at the end of life, supporting the theoretical concepts with the examples of real clinical situations (such as the paradigmatic heart failure management, that has been frequently mentioned).
Ethics related to the appropriate intensive care delivery is out of the purpose of the paper and should be probably restricted to few basic concepts, while ethics of palliative care is extensively addressed and should remain as it is. With regards to lines 337-340, where the Author claims the need of better defining CICU admittance criteria, it is in my opinion obscure the concept of “proportionality of treatments ” as well as table 2, both deserving further clarification.
Response: The chapters Epidemiology and Changes in the CICU in the third millennium were downsized.
The structure of the pargraphs is changed, according to the Reviewer suggestions; a complete definition of PC of WHO is inserted.
I think the best examples of real clinical situations are those concerning the deactivation of ICD and LVAD, fully analyzed in the parargraph dedicated to Withholding/withdrawing treatments in the CICU.
However some mention of real experience is added in the text.
I'd like to outline it is difficult to include clinical examples in a review.
A more detailed clarification about the concepts of appropriateness and proportionality of treatment were added.
Finally a short text about palliative sedation was inserted.
Round 2
Reviewer 2 Report
The Authors have provided the final version of the paper with the revision and corrections required. In this way I believe that they have further improved the quality of the paper